# Dual-Band Wide-Angle Reflective Circular Polarization Converter with Orthogonal Polarization Modes

**DOI:** 10.3390/s22249728

**Published:** 2022-12-12

**Authors:** Bianmei Zhang, Chenghui Zhu, Ran Zhang, Xiaofan Yang, Ye Wang, Xiaoming Liu

**Affiliations:** 1School of Physics and Electronic Information, Anhui Normal University, Wuhu 241002, China; 2School of Electrical and Electronic Engineering, Anhui Institute of Information Technology, Wuhu 241002, China; 3The State Key Laboratory of Complex Electromagnetic Environment Effects on Electronic and Information System, Luoyang 471004, China; 4Wuhu Ceprei Information Industry Technology Research Institute, Wuhu 241002, China

**Keywords:** metasurface, dual-band, circular polarization conversion, angular stability

## Abstract

Herein a dual-band wide-angle reflective circular polarization converter, based on a metasurface was developed. The unit cell is composed of a split square ring and a nested square patch. The split square ring plays the role of creating polarization conversion. The square patch is useful for improving the quality of axial ratio. It was verified that the structure could transform the *x*-polarized incident wave into left-hand circular polarization in the lower frequency band, and to right-hand circular polarization in the higher frequency band. For *y*-polarized incidence, the transformation has orthogonal modes to that for *x*-polarized incidence. Moreover, the 3 dB axial ratio takes place in the ranges of 8.42–12.32 GHz and 18.74–29.73 GHz, corresponding to a relative bandwidth of 37.61%, and 45.35%, respectively. In addition, the polarization conversion efficiency is greater than 99% in the ranges of 8.65–11.83 GHz and 19.55–29.36 GHz. Furthermore, for oblique incidence, the axial ratio remains stable, even at 50° incidence, for the lower frequency band. Lastly, a prototype is fabricated and measured for experimental verification. The measured and simulated results were in good agreement. Compared with other designs in the literature, the proposed converter operates with good performance in dual-band, with high-efficiency, and with angular stability.

## 1. Introduction

Metamaterials are artificial materials that can achieve many exotic properties of electromagnetic (EM) waves, such as anomalous refraction/reflection [1], photonic spin hall effects [2], focusing [3], vortex generation [4], and asymmetric transmission [5]. Due to their arbitrary values of permittivity and permeability, they can manipulate EM waves in an unprecedented manner. Recently, metasurfaces (MSs), as a two-dimensional metamaterial with subwavelength periodic structures, have attracted great interest as alternatives for controlling EM waves [6,7]. Compared with three-dimensional (3D) bulky metamaterials, MSs have the unequalled advantages of low loss, easy fabrication and integration. Based on this, many different types of MSs have been proposed, such as coding metasurface [8,9], gradient metasurface [10], and polarization conversion metasurface (PCM) [11]. In particular, polarization converters, based on PCM, have strong design capabilities and functional customization, which can realize different functions for different unit structures, materials and layouts, so it provides a feasible solution to manipulating polarization states in an effective and convenient way [12,13].

A polarization converter is one of the most important devices for polarization manipulation. Applications of polarization converters have been deployed in antenna radome [14], remote sensing [15], radar cross section (RCS) [16] and measurement and communication systems [17]. A range of polarization conversions are required in these applications. On the basis of MS technology, different types of polarization converters have been intensively studied to realize diverse functionalities, including linear to linear [18], linear to circular [19], circular to circular [8,9], and multifunctional polarization conversion [20,21]. Especially in regard to linear to circular polarization conversion, they have attracted considerable attention for sensor and satellite communication systems. In general, there are two types of MS configuration for circular polarization manipulation, namely, the transmission mode [22] and the reflection mode [23]. For the transmission mode, broadband polarization converters are regularly achieved with stacked multilayer structures [24,25,26,27], although the complex structures limit their applications in practice. In Ref. [24], a transmissive structure, consisting of capacitive patches and inductive wire, was introduced to realize circular polarization conversion, but the design only had a narrow operational band. To expand the bandwidth, many transmissive polarization converters were developed, such as cross-shaped patches [25], split-ring resonators [26,27], and the Jerusalem cross pattern [28]. However, most of these cannot operate at broadband, with wide-angle, and at high-efficiency, simultaneously. Moreover, some structures may suffer from high insertion loss. Therefore, many reflective polarization converters were designed and presented in Refs. [29,30,31,32,33,34,35,36,37,38,39]. It can be found that these structures in reflection mode are of great importance to obtain desirable EM properties, compared to the transmission mode.

For example, a reflective polarization converter was presented, based on W-shaped resonators, in Ref. [29]. The design could convert a linearly polarized incident wave into a circularly polarized wave with a bandwidth of 13%. The H-shaped structure in [30] achieved broadband circular polarization conversion by adjusting parameters. However, both designs only operated at normal incidence and narrow operational bands. Further efforts were made to broaden the bandwidth and the angular stability [31,32]. In Ref. [31], by virtue of metallic vias and multireflection, the operation bandwidth of the single band increased to 30%, with 20° angular stability. Another design [32] using polygon array also presented single band operation, providing 46% wide axial ratio bandwidth and 30° angular stability. 

As well as single band and broadband polarization converters, dual-band polarization converters have also been heavily investigated [33,34,35,36]. Fartookzadeh et al. [33] introduced a dual-band circular polarization converter based on anisotropic impedance surfaces. The structure could provide 39° angular stability, but it was very unstable, due to the existence of air gaps between the layers. A single substrate was used in Ref. [35], demonstrating 29%/11% bandwidth for −3 dB axial ratio over dual-band operation. However, the angular stability of this design was limited to 15°. Another design, in [36], also operated with a single substrate, which could enhance the angular stability to 30°; however, the dual-band operations were not wide enough, being less than 7%. In Ref. [37], a low-profile broadband polarization converter was designed with 84% bandwidth for the first band. However, this design showed low angular stability, and the incident angle was less than 15°. It can be observed that research on polarization conversion using MSs to design polarization converters still has great exploration potential.

In this work, a dual-band reflective circular polarization converter with orthogonal polarization modes was developed. It is composed of a single layer of square patch and square ring with two splits at the opposite corners. The split square ring creates polarization conversion, and the square patch is useful for improving the quality of the axial ratio. It is shown, in this work, that the structure could transform the *x*-polarized incident wave into right-hand circular polarization (RHCP) at 8.42–12.32 GHz and left-hand circular (LHCP) at 18.74–29.73 GHz. Moreover, for *y*-polarized incidence, the transformation had orthogonal modes to that for *x*-polarized incidence. In addition, this structure exhibited 50° angular stability for −3 dB axial ratio in the lower frequency band. To validate the design, a prototype was fabricated and measured. The measured results were well in agreement with the simulated ones. Compared with other polarization converters, the proposed converter exhibits superiorities of high-efficiency, dual-band operation and angular stability.

## 2. Analysis of Operation Principle

This section is devoted to a description of circular polarization conversion. An incident wave ${\vec{E}}^{i}$, can always be decomposed into two orthogonal components along the *x*-axis ($E_{x}^{i}$) and *y*-axis ($E_{y}^{i}$). Considering that a polarization converter is an anisotropic structure, when a linear polarized wave is incident on it, the relationships between the incident and reflected EM waves can be expressed as in [19]:(1)$$\left[\begin{array}{l} E_{x}^{r}\\ E_{y}^{r} \end{array}\right]=\left[\begin{array}{cc} r_{xx} & r_{xy}\\ r_{yx} & r_{yy} \end{array}\right]\left[\begin{array}{l} E_{x}^{i}\\ E_{y}^{i} \end{array}\right]=R_{lin}\left[\begin{array}{l} E_{x}^{i}\\ E_{y}^{i} \end{array}\right],$$
wherein the first and second subscripts, *I* and *j*, of $r_{ij}=E_{i}^{r}/E_{j}^{i}$ correspond to the polarized states of the reflected and incident waves, respectively.

Suppose an *x*-polarized wave is propagating along the +*z* direction, then the reflected wave can be described as in [29,31]:(2)$${\vec{E}}_{total}^{r}=r_{yx}E_{x}^{i}e^{-(jkz+\varphi_{yx})}\vec{y}+r_{xx}E_{x}^{i}e^{-(jkz+\varphi_{xx})}\vec{x},$$
where $r_{xx}$ and $r_{yx}$ represent the amplitudes of co-polarization and cross-polarization, respectively. The values $\varphi_{xx}$ and $\varphi_{yx}$ denote their corresponding phases.

If the amplitudes and phases between $r_{xx}$ and $r_{yx}$ satisfy the conditions of:(3)$$\left|r_{xx}\right|=\left|r_{yx}\right|,\Delta \varphi_{xy}=\varphi_{xx}-\varphi_{yx}=2k\pi +\frac{\pi }{2},$$
the reflected wave is in circular polarization, as shown in Figure 1a, where *k* is an integer. Similarly, the conditions are also valid for *y*-polarized incidence.

Since the structure always undergoes dielectric and conductor losses during the polarization conversion, the energy conversion ratio (*ECR*) is defined to characterize energy loss of the incident wave after hitting the polarization converter, which can usually be written as [38]:(4)$$ECR={\left|r_{xj}\right|}^{2}+{\left|r_{yj}\right|}^{2}(j=x,y).$$

Considering the reflected wave from the polarization converter is elliptical polarization in most cases, the reflected wave can be considered as a circular polarization one, when the axial ratio (*AR*) is lower than 3 dB. Thus, the *AR* is applied to evaluate this polarization conversion property, which can be calculated by [21,32]
(5)$$\left\{ \begin{array}{l} AR={\left(\frac{{\left|r_{xj}\right|}^{2}+{\left|r_{yj}\right|}^{2}+\sqrt{a}}{{\left|r_{xj}\right|}^{2}+{\left|r_{yj}\right|}^{2}-\sqrt{a}}\right)}^{1/2}\\ a={\left|r_{xj}\right|}^{4}+{\left|r_{yj}\right|}^{4}+2{\left|r_{xj}\right|}^{2}{\left|r_{yj}\right|}^{2}\cos\left(2\Delta \varphi_{xy}\right) \end{array}\right.,(j=x,y).$$

Furthermore, to demonstrate the high polarization conversion efficiency (*PCE*) of the reflected wave, the reflection coefficients of linear to circular polarization conversion can be expressed as [39]:(6)$$\left\{ \begin{array}{l} r_{LHCP-j}=\sqrt{2}(r_{yj}+ir_{xj})/2\\ r_{RHCP-j}=\sqrt{2}(r_{yj}-ir_{xj})/2 \end{array}\right.,(j=x,y).$$

So that the *PCE* can be derived using the following equation:(7)$$PCE=\frac{{\left|r_{RHCP-j}\right|}^{2}}{{\left|r_{RHCP-j}\right|}^{2}+{\left|r_{LHCP-j}\right|}^{2}}.$$

## 3. Design and Simulation

### 3.1. Linear to Circular Polarization Conversion

The proposed polarization converter and associated parameters are shown in Figure 1b. The unit cell was formed by two copper layers, separated by a dielectric layer, with a relative dielectric permittivity of 2.65 and a loss tangent of 0.002. The top metallic layer of the unit cell is composed of a square patch and square ring with two splits at the opposite corners, while the back layer is a purely metallic plane with an electric conductivity of σ = 5.8 × 10^7^ S/m and a thickness of 0.035 mm. The split square ring is used for creating circular polarization conversion, and the conversion quality can be improved by adjusting the dimension of the nested square patch. By optimizing the geometry parameters, dual-band circular polarization is generated by resonant frequencies. The resulting design parameters of the structure were chosen to be as follows: *p* = 5.8 mm, *h* = 2.1 mm, *l*_0_ = 5.34 mm, *l*_1_ = 0.9 mm, *w* = 0.25 mm and *g* = 0.66 mm.

To numerically investigate the performance of the linear to circular polarization conversion, the design was simulated using Ansoft HFSS, where the Floquet port was placed in the +*z* direction with the wave vector along the −*z* direction and Master/Slave boundaries in +*x*/−*x* and +*y*/−*y* directions, respectively. Since the unit cell contained two parts, the unique function of each subsection for the structure was investigated, as illustrated in Figure 2 and Figure 3. It can be seen from Figure 2 that the split square ring could stimulate two resonances, which realized circular polarization conversion in two separate frequency bands. It is worth noting that the split ring, as resonator, is a common way to create multiple resonances, so that several bands may be created. By adding the square patch, the amplitude conditions were considerably improved, resulting in better circular polarization conversion. As shown in Figure 3a, the amplitudes of $r_{xx}$ and $r_{yx}$ were nearly equal in the ranges of 8.42–12.32 GHz and 18.74–29.73 GHz.

The phase difference $\Delta \varphi_{xy}$ is plotted in Figure 3b. It can be seen that $\Delta \varphi_{xy}$ was nearly equal to an odd multiple of π/2 in the whole frequency region, indicating that circular polarizations were created by the *x*-polarized incident wave in the frequency ranges of 8.42–12.32 GHz and 18.74–29.73 GHz. Accordingly, the *AR* is plotted in Figure 3c, and it can clearly be seen that the *AR* was less than 3 dB at 8.42–12.32 GHz and 18.74–29.73 GHz, corresponding to a relative bandwidth of 37.61%, and 45.35%, respectively. Moreover, the *AR* remained below 1 dB in the frequency ranges of 8.84–11.46 GHz and 20.23–26.49 GHz. Even the *AR* was lower than 0.5 dB in the frequency regions of 9.01–11.16 GHz and 20.85–25.41 GHz. This result implied that nearly perfect circular polarizations had been achieved over the dual-band operation regions. 

To assess the handedness of the reflected wave, the normalized ellipticity is frequently expressed as that in Ref [17]:(8)$$e=\frac{2r_{yx}r_{xx}\sin\Delta \varphi_{xy}}{r_{yx}^{2}+r_{xx}^{2}}.$$

The calculated *e* of the proposed converter is shown in Figure 4a. It is seen that the *e* approached −1 in the range of 8.42–12.32 GHz and it approached +1 in the region of 18.74–29.73 GHz. This was good evidence showing that, for normal incidence, the structure could convert the *x*-polarized wave into LHCP in the lower band and into RHCP in the higher band. It was noted that the same phenomenon could be created for *y*-polarized incidence, but with orthogonal polarization modes at each band. This was reasonable, due to the anisotropic property of the structure. The calculated ECRs, using Equation (4) are shown in Figure 4b, where it can be observed that almost 97% of the energy was reflected within two operational bands. Such a result indicated that, for both *x*-polarization and *y*-polarization incidences, the structure had a satisfactory low-loss property over the dual-band operational regions.

Next, in order to prove that the proposed converter had a high *PCE*, the reflection coefficients and *PCE* of the reflected circular polarization wave are shown in Figure 5. It can be observed from Figure 5a that the amplitudes of $r_{RHCP-x}$ and $r_{LHCP-x}$ were very close to 0 dB in the ranges of 8.42–12.32 GHz and 18.74–29.73 GHz. Furthermore, by examining the *PCE* in Figure 5b, it was clearly found that the *PCE* was greater than 99% in the regions of 8.65–11.83 GHz and 19.55–29.36 GHz, occupying 81.54% and 89.26% of the 3 dB *AR* bandwidth, respectively. It was apparent that the proposed polarization converter could achieve circular polarization conversion with high efficiency in the two frequency regions.

A circularly polarized incident wave can always be decomposed into two perpendicular linear polarized waves with equal amplitude and $\pm 90^{\circ }$ phase difference. Thus, the reflection matrix $R_{cir}$ at the circularly polarized incidence could be derived from $R_{lin}$ at the linearly polarized incidence in Equation (1), and could be expressed as [39]:(9)$$R_{cir}=\left[\begin{array}{cc} r_{++} & r_{+-}\\ r_{-+} & r_{--} \end{array}\right]=\frac{1}{2}\left[\begin{array}{cc} r_{xx}-r_{yy}-i(r_{xy}+r_{yx}) & r_{xx}+r_{yy}+i(r_{xy}-r_{yx})\\ r_{xx}+r_{yy}-i(r_{xy}-r_{yx}) & r_{xx}-r_{yy}+i(r_{xy}+r_{yx}) \end{array}\right]$$

Due to the symmetry of the polarization converter structure, one had $r_{xx}=r_{yy}$, $r_{yx}=r_{xy}$. Then, the reflection matrix $R_{cir}$ could be simplified as:(10)$$R_{cir}=\left[\begin{array}{cc} r_{++} & r_{+-}\\ r_{-+} & r_{--} \end{array}\right]=\left[\begin{array}{cc} ir_{xy} & r_{yy}\\ r_{yy} & -ir_{xy} \end{array}\right]=\left[\begin{array}{cc} ir_{yx} & r_{xx}\\ r_{xx} & -ir_{yx} \end{array}\right]$$

Therefore, when the proposed structure was considered as a circular polarization converter, the reflected wave would be in linear polarization for a circularly polarized incident. However, when the proposed structure was considered as a linear polarization converter, a circular polarization would be formed for the reflected wave. Moreover, for circularly polarized incidence, the calculated *AR* of the proposed converter is shown in Figure 6. It can be seen that the *AR* was greater than 12 dB in the frequency ranges of 8.42–12.32 GHz and 18.74–29.73 GHz. However, the *AR* was less than 3 dB in the frequency region of 13.85–15.14 GHz. This was good evidence, showing that two linear polarizations were generated in the frequency ranges of 8.42–12.32 GHz and 18.74–29.73 GHz, while a circular polarization formed in the frequency region of 13.85–15.14 GHz.

### 3.2. Surface Current Distributions

The proposed polarization converter exhibited a property of dual-band operation with orthogonal polarization modes. To better understand the mechanism behind the polarization conversion, the surface current distributions were presented at their resonance frequencies of 10 and 23 GHz, as shown in Figure 7. The surface current was concentrated on the split square ring surface, and flowed along the *x*-direction and *y*-direction. Moreover, it can be seen from Figure 7a,b that the current distributions were in opposite directions at their resonance frequencies of 10 and 23 GHz. Such a property indicated that circular polarizations with orthogonal polarization modes could be generated for two separate frequency bands, when the phase condition of the two orthogonal directions was satisfied.

### 3.3. Parametric Analysis

To investigate the sensitivity of *AR* response on different parts of the unit cell, a parametric analysis was conducted against some critical unit cell dimensions, as shown in Figure 8. Since the parameters *l*_0_, *l*_1_, *w* and *g* of the unit cell were less sensitive for variations of dimension, a 100 µm simulation step was applied for this design. Figure 8a depicts the variations of *AR* for different lengths *l*_0_. It can be observed that the operation region of 3 dB *AR* shifted slightly towards a lower frequency as *l*_0_ varied from 5.24 mm to 5.44 mm. A similar phenomenon could be observed for the parameter *l*_1_ in Figure 8b. It should be noted that the dimensions of the metallic patch determined the inductance values for the proposed converter, further resulting in a shift of the operational region to lower or higher frequencies. Moreover, the *AR* response for different widths, *w*, is also plotted in Figure 8c. It can be seen that the working bandwidth for 3 dB *AR* decreased gradually in the lower frequency band when the parameter *w* changed from 0.15 mm to 0.35 mm, while the operating bandwidth effectively increased for the higher frequency band. Considering the dual-band operational regions and conversion performances, the final dimensions of the parameters *l*_0_, *l*_1_, *w* and *g* were determined, as shown in Figure 1b. 

Another important parameter in the polarization converter was the corner width, *g*, of the unit cell. The influence of the parameter *g* on *AR* is shown in Figure 8d. It is seen that, for a smaller dimension, i.e., *g* = 0.56 mm, the capacitance between the corner had a larger value, thus, shifting the 3 dB *AR* response to the lower frequencies. However, by increasing *g* in the range of 0.56–0.76 mm, the 3 dB *AR* curves shifted towards higher frequencies. Consequently, the dual-band operational regions of the proposed polarization converter could be effectively manipulated by adjusting the dimensions of the unit cell.

### 3.4. Angular Stability

The incident angular stability of the linear to circular polarization conversion is important in practical applications. For different incident angles, the simulated *AR* results of the reflected wave are shown in Figure 9. From Figure 9a, it is clearly observed that the bandwidth of 1 dB *AR* gradually decreased with the incident angle changing from 0° to 50°, while the bandwidth for 3 dB *AR* was less sensitive to the oblique incident angle. Moreover, the 3 dB *AR* bandwidth widened as the incident angle increased, and the relative bandwidth increased to 41.75% in the frequency range of 8.32–12.71 GHz when the incident angle reached 50°. These results demonstrated that the structure had good angular stability in the lower band, even though the incident angle rose to 50°. 

In addition, the ARs from the higher frequency region were also plotted for different incident angles, as shown in Figure 9b. The 3 dB *AR* bandwidth remained relatively stable when the incident angle was in the range of 0°–20°, while some undesired bumps appeared around 23 GHz. Even worse, the bandwidth rapidly reduced with the increase of the incident angle in the range of 25°–30°, especially in the higher frequencies, which might be attributed to the effect of the destructive interference of the surface from the polarization converter. It was clear that the angular stability of the polarization converter in the lower band was better than that in the higher band. This might be explained by the fact that the converter had a smaller dimension in the lower band, resulting in less sensitivity for the oblique incidence.

## 4. Experimental Results

To experimentally validate the design, a prototype was fabricated by using standard print circuit board (PCB) technology, as shown in Figure 10. The sample consisted of 41 × 41 unit cells, etched on an F4B substrate with a total size of 285 mm × 285 mm. By using an industrial microscope (SuperEyes), the accuracy of the fabricated sample was examined. It can be seen that the fabricated accuracy remained within 10 μm, which was very beneficial to improving the bandwidth and angular stability of this design.

The structural diagram of the measurement setup is shown in Figure 11. Two standard-gain horn antennas, one as the transmitter and the other as the receiver, were connected to the vector network analyzer (Ceyear AV3672D). The sample was arranged in the far field area of the antenna and surrounded by radar absorbing materials (RAM), which were applied to avoid diffraction and spillover at the edges. For $r_{xx}$ measurement, both transmitter and receiver horn antennas were placed along the same orientation, while for the $r_{yx}$ measurement the receiver horn antenna was rotated by 90°. Moreover, to calibrate the system, a metal sheet of the same size as the sample was also measured, so that the *AR* could be obtained.

Figure 12 shows the measured and simulated *AR* results for different oblique incidences. It is seen that the measured results were satisfactorily consistent with the simulated ones. It has to be mentioned that the incident angle of 5° was applied to mimic normal incidence for the measurement system. This was because of the physical limitation between the transmitter and receiver horns. From Figure 12, it can be observed that at nearly normal incidence the *AR* of the reflected wave was below 3 dB in the frequency ranges of 8.04–12.06 GHz and 18.18–29.4 GHz. Moreover, for oblique incidence, the *AR* remained stable in the lower band, but deteriorated in the higher band, as the incident angle increased. In the lower band, the 3 dB *AR* frequency range was 8.28–12.06 GHz when the incident angle changed from 5° to 50°. In the higher band, although some undesired bumps appeared in the *AR* curves over the operational band, the *AR* still remained below 3 dB in the frequency ranges of 18.9–23.1 GHz and 23.94–27.36 GHz (except for some undesired bumps around 23.5 GHz) as the incident angle varied in the range of 5°–20°. This was reasonable, since the structure had a smaller dimension in the lower band, resulting in less sensitivity to the incident angle. However, there were also some slight discrepancies between simulation and measurement. This could have been caused by fabrication and measurement errors, such as misalignment of the horn antennas and the finite number of unit cells in the MS construct.

A comparison between the proposed polarization converter and the reported literature is presented in Table 1. It can be observed that the bandwidth of this design was much wider than the designs in Refs. [26,27,28,29,30,31,33,34,35,36,38]. Even if there were solutions, such as those in [25,32,37], that outperformed the one proposed in this work, these structures were developed using more metallic layers or a more complex pattern, and were less preferable in view of angular stability. In Refs. [24,26,27,34,38], these structures provided a good stability response with the oblique incidence angle, while the operation bands were limited. In the comparison, it can be found that the proposed converter provided a unique combination of dual-band operation, a thin profile, and stable response to incident angle.

In consideration of the attractive merits of this design, potential applications can be envisaged in antenna radome, remote sensors, and dual-band communication systems etc. Particularly for dual-band polarization isolation, it could be a promising design. 

## 5. Conclusions

A dual-band wide-angle reflective circular polarization converter was developed. The unit cell consisted of a single layer of square patch and square ring with two splits at the opposite corners. It could transform a given linear polarized wave into a circular polarized wave with orthogonal polarization modes in two separate frequency bands. The simulated results showed that the *AR* remained below 3 dB in the frequency ranges of 8.42–12.32 GHz and 18.74–29.73 GHz, corresponding to a relative bandwidth of 37.61%, and 45.35%, respectively. Moreover, for oblique incidence, the structure provided high angular stability with the oblique angle up to 50° in the lower band. To validate the design, a prototype was fabricated and measured. The measured results were consistent with the simulated ones. 

## Figures and Tables

**Figure 1 sensors-22-09728-f001:**
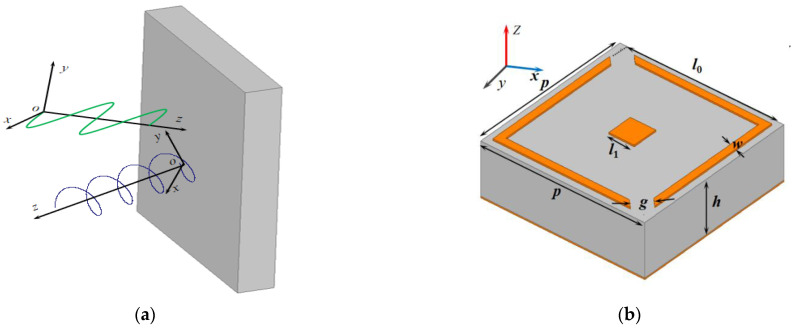
(**a**) The illustration of linear to circular polarization conversion; (**b**) The unit cell and its parameters.

**Figure 2 sensors-22-09728-f002:**
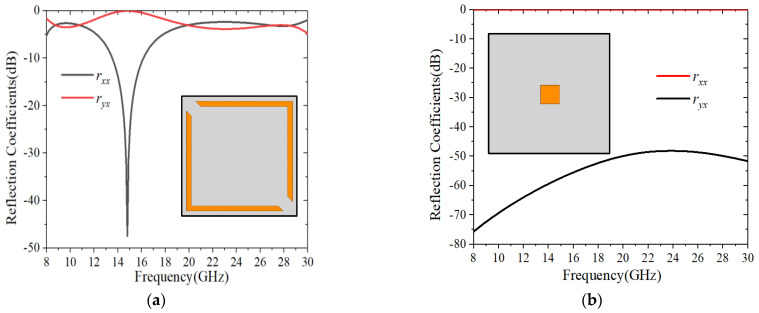
Reflection coefficients for: (**a**) Split square ring array; (**b**) Square patch array.

**Figure 3 sensors-22-09728-f003:**
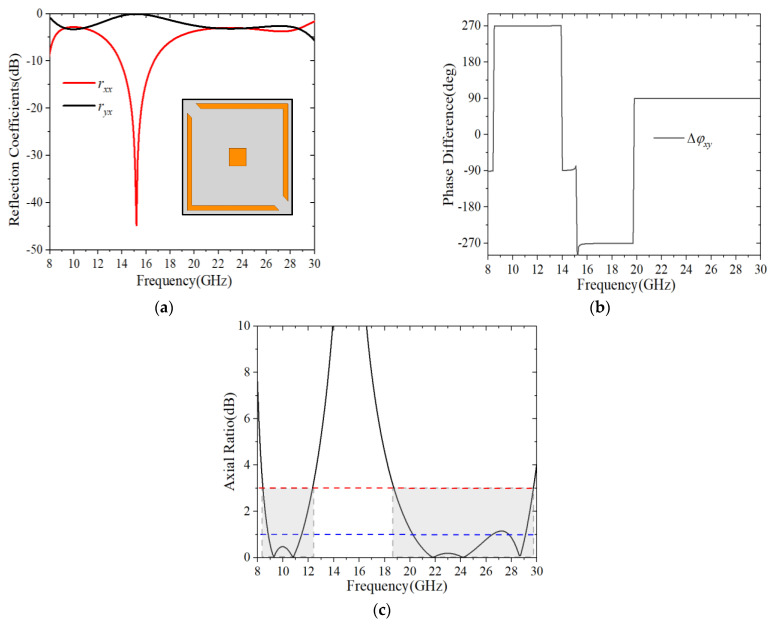
Simulation results for normal *x*-polarized incidence. (**a**) Amplitude; (**b**) Phase difference; (**c**) Axial ratio.

**Figure 4 sensors-22-09728-f004:**
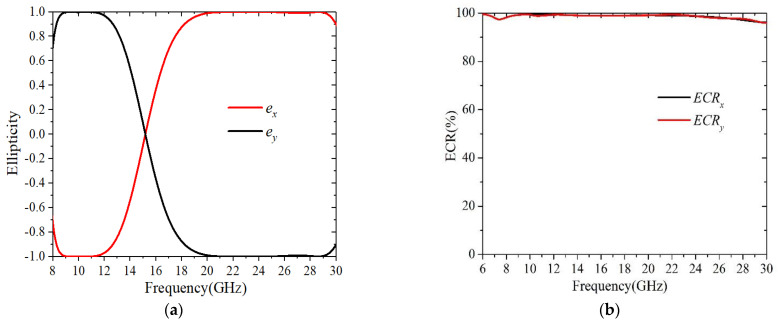
(**a**) Normalized ellipticity; (**b**) Energy conversion ratio.

**Figure 5 sensors-22-09728-f005:**
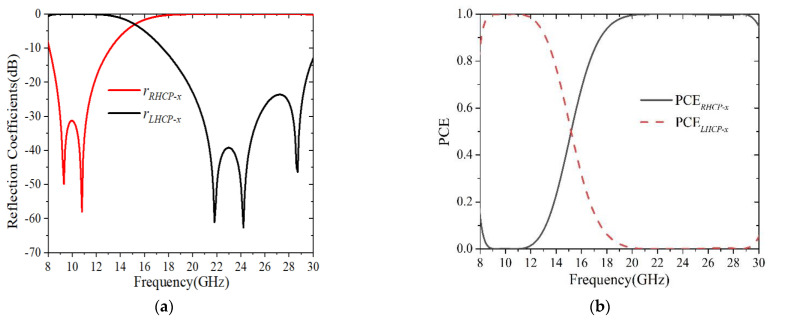
(**a**) Reflection coefficients of circular polarization conversion; (**b**) *PCE* of circular polarization conversion.

**Figure 6 sensors-22-09728-f006:**
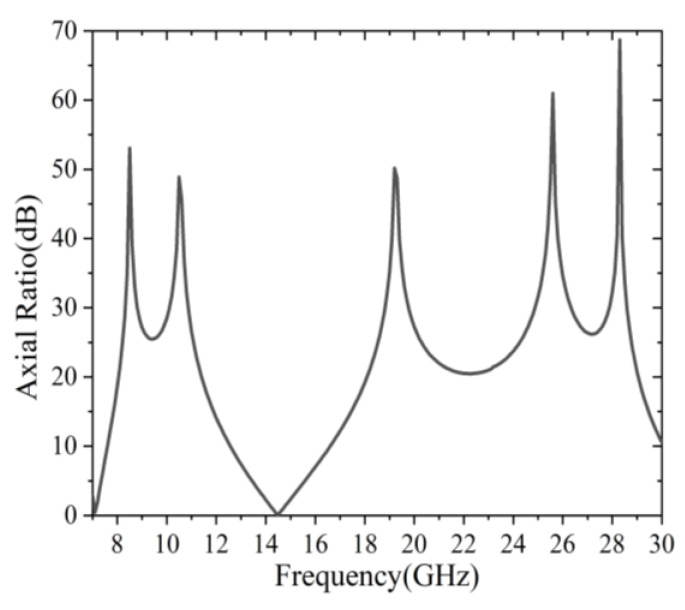
Simulated axial ratio for normal circular polarized incidence.

**Figure 7 sensors-22-09728-f007:**
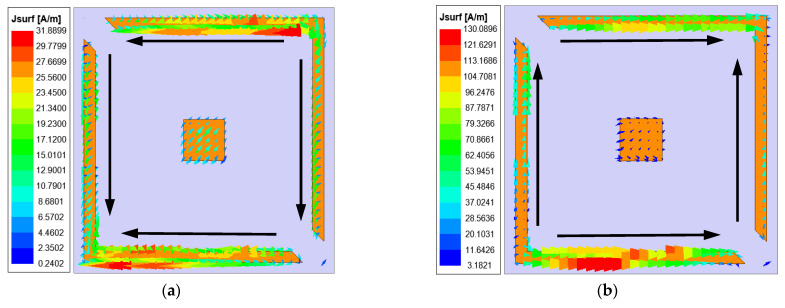
Surface current distributions at their resonance frequencies of (**a**) 10 GHz; (**b**) 23 GHz.

**Figure 8 sensors-22-09728-f008:**
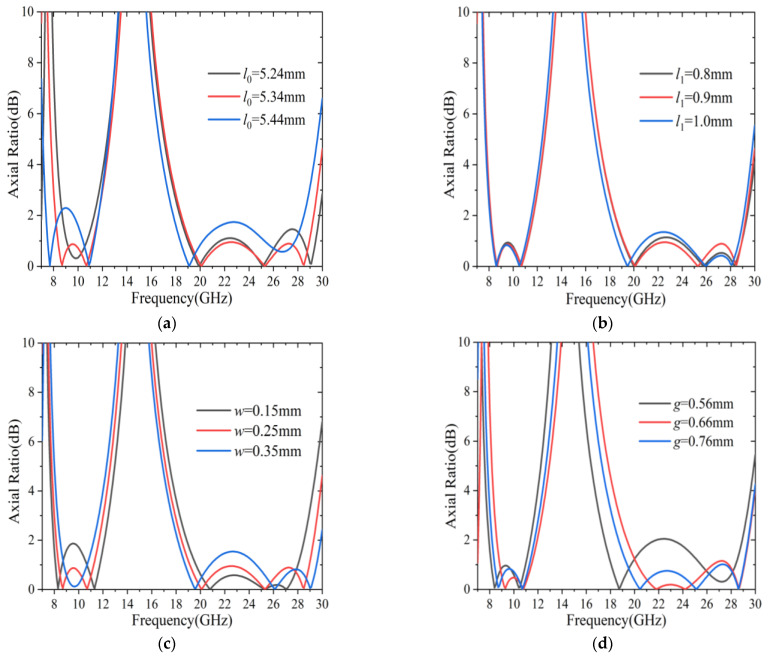
The axis ratio variation dependence on different parameters of the unit cell: (**a**) *l*_0_; (**b**) *l*_1_; (**c**) *w*; (**d**) *g*.

**Figure 9 sensors-22-09728-f009:**
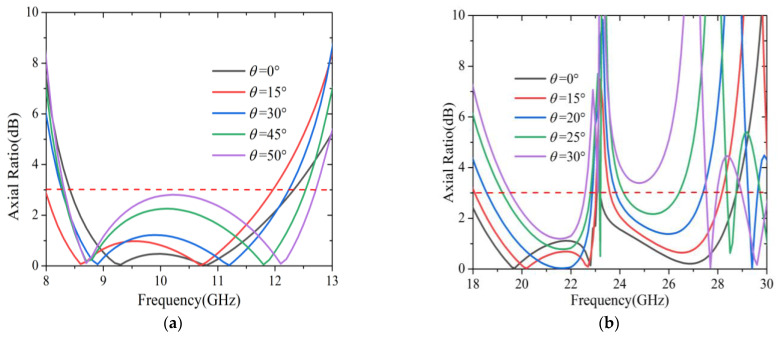
The axis ratio under different oblique incidence for: (**a**) The lower band; (**b**) The higher band.

**Figure 10 sensors-22-09728-f010:**
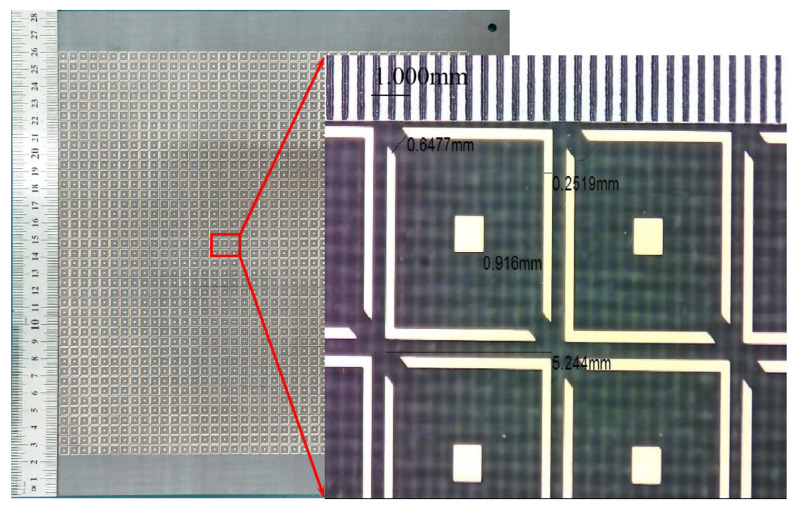
Photograph of the fabricated sample.

**Figure 11 sensors-22-09728-f011:**
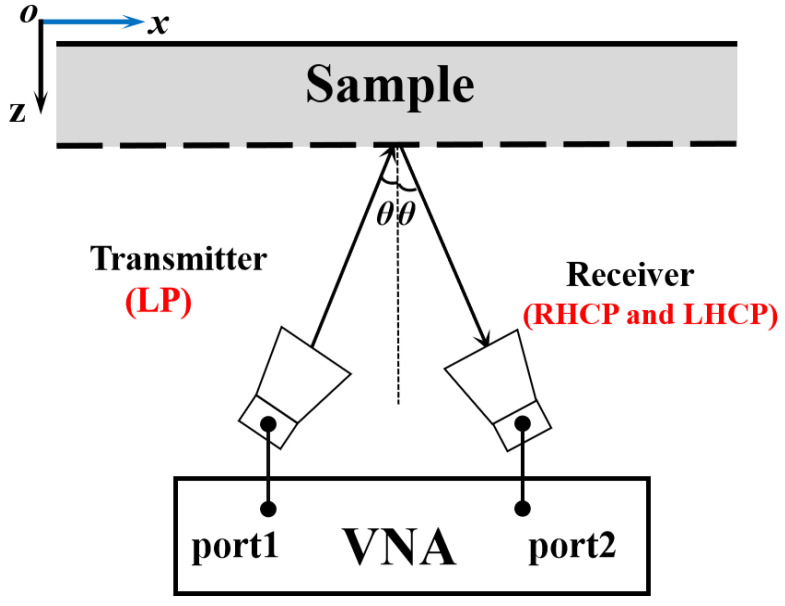
Structural diagram of the measurement setup.

**Figure 12 sensors-22-09728-f012:**
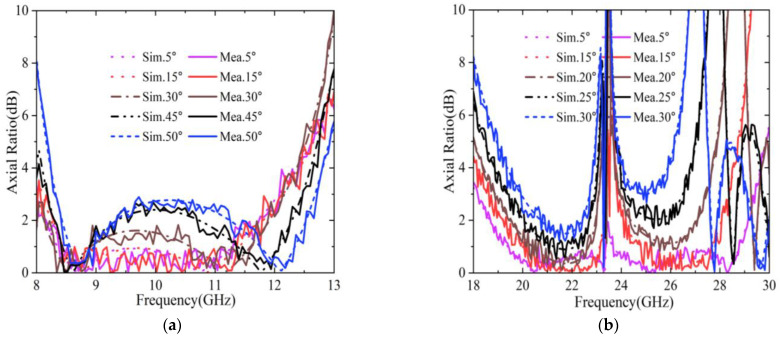
Measured and simulated axial ratios in: (**a**) The lower band; (**b**) The higher band.

**Table 1 sensors-22-09728-t001:** Performance comparison of polarization converters in the literature.

Ref.	Type	Frequency (GHz)	Metallic Layers	Structure Thickness	Unit Cell Size	3 dB *AR* Bandwidth	Angular Stability
[24]	transmission	8–12	≥3	0.144 λ_L_	0.12 λ_L_ × 0.12 λ_L_	40%	45°
[25]	transmission	5.15–11.2	≥3	0.103 λ_L_	0.258 λ_L_ × 0.258 λ_L_	74%	20°
[26]	transmission	9.05–9.65, 12.55–13.1	≥3	0.06 λ_L_	0.272 λ_L_ × 0.272 λ_L_	6.42%, 4.29%	55°
[27]	transmission	19.7–20.2, 29.5–30	≥3	0.069 λ_L_	0.348 λ_L_ × 0.348 λ_L_	2.51%, 1.68%	45°
[28]	transmission	16.28–20.82, 26.94–30.13	2	0.085 λ_L_	0.217 λ_L_ × 0.217 λ_L_	24.47%, 11.18%	20°
[29]	reflection	13.7–15.6	2	0.142 λ_L_	0.37 λ_L_ × 0.37 λ_L_	12.97%	0°
[30]	reflection	14.95–17.35	2	0.15 λ_L_	0.449 λ_L_ × 449 λ_L_	14.86%	0°
[31]	reflection	6.8–9.2	2	0.157 λ_L_	0.204 λ_L_ × 0.204 λ_L_	30%	20°
[32]	reflection	7.62–12.16	2	0.041 λ_L_	0.191 λ_L_ × 0.191 λ_L_	45.9%	30°
[33]	reflection	1.9–2.3, 7.9–8.3	2	-	-	19.04%, 4.94%	39°
[34]	reflection	20.2–21.2, 29.5–30.8	≥3	0.068 λ_L_	0.303 λ_L_ × 0.303 λ_L_	4.83%, 4.31%	60°
[35]	reflection	15–20, 27–30	2	0.076 λ_L_	0.489 λ_L_ × 0.489 λ_L_	28.57%, 10.53%	15°
[36]	reflection	11.7–12.5, 17.3–18.1	2	0.059 λ_L_	0.156 λ_L_ × 0.312 λ_L_	6.61%, 4.52%	30°
[37]	reflection	6.87–16.83, 19.72–23.64	2	0.057 λ_L_	0.192 λ_L_ × 0.192 λ_L_	84.05%, 18.08%	15°
[38]	reflection	29–41.5, 52.5–61.5	2	0.077 λ_L_	0.193 λ_L_ × 0.193 λ_L_	35.46%, 15.79%	45°
This work	reflection	8.42–12.32, 18.74 –29.73	2	0.059 λ_L_	0.163 λ_L_ × 0.163 λ_L_	37.61%, 45.35%	50°

## Data Availability

Not applicable.

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
