# Peer review of "Dual-Band Wide-Angle Reflective Circular Polarization Converter with Orthogonal Polarization Modes"

_sensors, 2022, doi:10.3390/s22249728_

Round 1

Reviewer 1 Report

Looking at the recent studies, a stronger circular polarization conversion was performed by taking the AR as -1 dB as a reference. AR is -3dB taken as reference in this study. Therefore, I believe that the design can be improved at the point of innovation. It is also important to evaluate the study for potential use applications. In addition, for a universal view of the study, it is necessary to give details of the literature covering all regions of the world.

Reviewer 2 Report

Authors have done interesting work related to the design of polarization conversion metasurface. The proposed work is operating in two bands. However, I have few questions related to the work

1. Please explain more about the novelty of the work done as I think there is numerous research paper present related to the design of PCM.

2. Please explain what will be the reflected wave polarization if the incident wave is circularly polarised.

3 Rest of the questions are mentioned in the PDF attached [please see highlighted part]

Round 2

Reviewer 2 Report

The authors have satisfactorily responded to all my queries.